## Replications

 

**Subject Category:**
Biology (whole organism)

behaviour/cognition/psychology

ageing, macaque, novelty

**Author for correspondence:**
Eliza Bliss-Moreau
e-mail: eblissmoreau@ucdavis.edu

# Interest in non-social novel stimuli as a function of age in rhesus monkeys

Eliza Bliss-Moreau[1] and Mark G. Baxter[2]

[1]Department of Psychology, California National Primate Research Center, University of California, Davis, CA, USA
[2]Nash Family Department of Neuroscience, Friedman Brain Institute, Mount Sinai School of Medicine, New York, NY, USA

EB-M, 0000-0002-0740-5612; MGB, 0000-0002-8907-0923

Human cognitive and affective life changes with healthy ageing; cognitive capacity declines while emotional life becomes more positive and social relationships are prioritized. This may reflect an awareness of limited lifetime unique to humans, leading to a greater interest in maintaining social relationships at the expense of the non-social world in the face of limited cognitive and physical resources. Alternately, fundamental biological processes common to other primate species may direct preferential interest in social stimuli with increasing age. Inspired by a recent study that described a sustained interest in social stimuli but diminished interest in non-social stimuli in aged Barbary macaques, we carried out a conceptual replication to test whether old rhesus monkeys lost interest in non-social stimuli. Male and female macaques (*Macaca mulatta*; N = 243) 4–30 years old were tested with a food puzzle outfitted with an activity monitor to evaluate their propensity to manipulate the puzzle in order to free a food reward. We found no indication that aged monkeys were less interested in the puzzle than young monkeys, nor were they less able to solve it.

## 1. Introduction

Social structure has a profound impact on human health (e.g. [1–4]) and so there is a need to understand the mechanisms by which social environment, affective processing, cognition and physical health interact with each other over the lifespan. The challenges associated with carrying out such mechanistic studies in humans have spurred interest in understanding how ageing impacts social and affective processes in non-human animals and the extent to which changes in those processes mirror those in humans [5–8]. Macaque monkeys, in particular, are well suited for studies of affective and cognitive ageing because of their sophisticated social abilities and behavioural repertoire, well-differentiated neocortex and neuroendocrine ageing processes which share similarities with humans.

**Figure 1.** Food retrieval puzzle. (*a*) Closed puzzle. (*b*) Puzzle bottom, showing interior tube which is shorter than the exterior tube and open on the bottom. (*c*) Puzzle top, showing an unwrapped activity monitor placed to demonstrate its positioning. Food reward is placed in the bottom around the interior tube, the top is screwed onto the bottom and then the puzzle is set in the cage with the bottom on the cage floor and the top facing up.

A fundamental question in carrying out these kinds of cross-species comparisons is whether there are evolutionarily conserved processes that may drive changes in social and affective behaviour as a function of age. The existence of such processes would smooth mechanistic comparisons between non-human primates and humans and would imply that changes in human behaviour in ageing are driven at least in part by mechanisms that do not require uniquely human abilities such as language or an awareness of one's own mortality as has been proposed as a mechanism in humans (e.g. [9,10]). A key recent finding in this regard was a report that aged Barbary macaques (*Macaca sylvanus*) lost interest in novel non-social stimuli—objects baited with food—but maintained interest in members of their own species with age [11]. This would suggest that prioritizing social relationships at the expense of interest in the non-social world, given limited physical and cognitive resources with increased age, may be related to more fundamental biological factors rather than a uniquely human awareness of the probable length of one's lifespan.

We were surprised by the finding that aged monkeys were less willing to manipulate a novel physical object to receive a food reward and less successful at freeing the food reward when they did [11], in light of our own experiences teaching aged monkeys to manipulate novel objects to receive food rewards as part of cognitive testing. Aged rhesus monkeys readily perform cognitive tasks that require them to touch and move physical objects of various shapes, sizes and colours in order to obtain rewards [12–17]. When memory demands are minimal, aged monkeys are able to perform at comparable levels to young monkeys. Countering this, and potentially consistent with the findings of Almeling and colleagues [11], is intriguing evidence (albeit from a small number of aged monkeys) that older monkeys tend to make fewer eye movements directed at novel visual stimuli after short delays between the initial presentation of the visual stimulus and a subsequent test phase [18].

Given these findings, we sought to evaluate age-related changes in monkeys' 'exploratory and problem solving' ([11], p. 1744) behaviour. We presented 243 rhesus monkeys (*Macaca mulatta*) each with a food retrieval puzzle filled with a reward (dried vegetable chips) (figure 1). Because the prior report [11] showed a decline in non-social interest (exploration) but not in social interest in aged macaques, our replication focused on whether we could reproduce the significant decline in non-social interest as previously described, rather than a non-effect of age on interest in social stimuli. We pre-registered our design on the Open Science Framework prior to data collection (https://osf.io/njqp4/). In our study, each puzzle was fit with an activity monitor in order to quantitatively index each animals' physical manipulation of the object during each of two 20-min test sessions conducted on two successive test days. Activity monitors recorded the maximum motion in the *X*-, *Y*- or *Z*-direction per 15 s time bin. The puzzle required physical manipulation—rotating the top over the bottom—in order to free the food reward. In order to directly compare our findings to those of Almeling and colleagues [11], we first evaluated the first 2 min of each test session followed by an evaluation of the entire 20 min period of each test day. This design allowed us to carry out comparable analyses to Almeling and colleagues [11] but also evaluate additional data in the event of different results from the primary analysis.

Our goal for the present study was to test competing hypotheses about the impact of age on object exploration and interest in non-social stimuli: whether interest would decline with age as reported by Almeling and colleagues [11] or whether we would see sustained interest with age, a hypothesis based on our previous experiences working with aged monkeys. Because of variations in study design, testing context and species (*M. mulatta* versus *M. sylvanus*), our study is a conceptual rather than a direct replication, with the intent of testing the same psychological construct by a similar technique (i.e. exploration of a physical

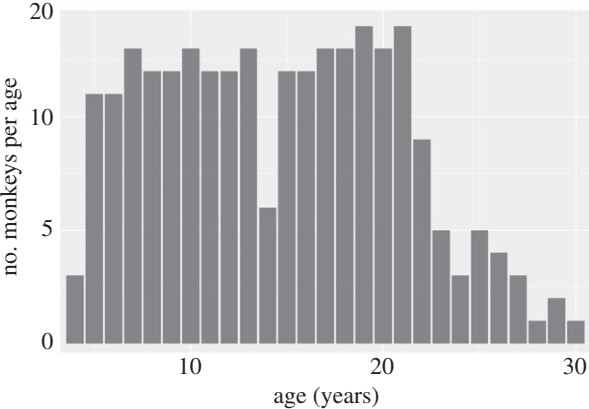

**Figure 2.** Subject distribution by age.

object to free a food reward) as it varies with the same independent variable (i.e. age). Given practical constraints of research with non-human primates that make exact replications difficult or impossible, in our view, such conceptual replications have value. Management practices, including social housing, differ significantly across facilities. Further, macaques as a genus are often considered in the psychological and neuroscience literatures as a unitary type. However, data from anthropological and field studies point to differences in their social behaviour and societal structures which may very well have an impact on psychological processes such as motivated action for rewards, as tested here. If the goal of replication science is to understand the extent to which a phenomenon is robust and emerges in multiple samples, then it is critical to test multiple species even if features like their housing are varied by and/or are constrained by different management practices. Further, because of the limited number of non-human primates (especially aged animals) that are available to participate in studies and the significant challenges associated with working with them, the vast majority of non-human primate studies are *never* replicated, either precisely or conceptually. The potential impact of this is exacerbated by typically small sample sizes. Thus, we see conceptual replication of this nature—where we test the same construct as in the original paper in a very large sample—as an important step forward in understanding the evolution of psychological processes.

# 2. Material and methods

All procedures were approved by the University of California, Davis Institutional Animal Care and Use Committee.

This study was pre-registered before data collection: osf.io/njqp4. The manuscript then received results-blind in-principle acceptance (IPA) at Royal Society Open Science. Following IPA, the accepted Stage 1 version of the manuscript, not including results and discussion, was pre-registered on osf.io/rnzs4. This secondary preregistration was performed after data analysis. Data are available in https://doi.org/10.5061/dryad.1bj133v.

## 2.1. Subjects

The final sample included $N = 243$ rhesus monkeys (*M. mulatta*; $N = 141$ females) living at the California National Primate Research Center in the indoor colony. Monkeys were housed in standard indoor caging (size based on their weight, as per National Institute of Health standards) either alone, or with social pair-mate either 8 h per day or 24 h per day based on social compatibility and their ability to share food. During testing, subjects who were socially housed were separated into their individual cages by closing a pairing door and then re-paired after testing. All animals were fed monkey chow twice daily, fresh produce multiple times per week and had access to daily enrichment (forage, toys) and water ad libitum. Subjects ranged in age from 4 to 30 (figure 2).

## 2.2. Food retrieval puzzle

The food retrieval puzzle (figure 1) was composed of PVC pipe fittings available at home supply stores and designed by staff in Behavioral Health Services at the California National Primate Research

Center. The puzzle includes a small piece of pipe on the interior that so that the monkey must rotate the object in space to free the food reward inside. We used commercially available dried vegetable chips as the food reward ('Veggie Relish' from LabDiet, St Louis, MO) sorted by hand to ensure that all pieces fit through the small interior pipe. Each puzzle was filled with 35 g of food reward on each test day.

A Philips Respironics ActiCal (Koninklijke Philips, The Netherlands) activity monitor was wrapped in Parafilm (Bemis NA, Neenah, WI), attached to the inside lid of the puzzle with a piece of 3M double-sided tape (3MVHB; 3M, Maplewood, MN) and covered with a small piece of duct tape. The monitors were programmed via their associated software (Actical, v. 2.12) to collect data in 15 s time bins. The technicians started the monitors recording, affixed the top and then immediately gently set them into each animals' cage. They recorded continuously until they were removed, just over 20 min later (dependent on how long it took to get the puzzles back from the monkeys). We included 20 min of data from the beginning of the trial. Upon completion of each test for each animal, the puzzles were removed from each animal's cage with the opening side up and placed immediately into a small re-sealable bag. The remaining mass of food reward was then computed. Monkeys were tested on two consecutive days.

Our dependent variables were both the number of active bins and total activity counts (representing the magnitude of activity) for the first 2 min of each test day, corresponding to the 2 min exposure used in [11], and for the entire 20 min period on each test day, as well as the total mass of food reward the animal was able to free from the puzzle. Finally, we calculated an 'efficiency' measure of how much food was recovered during the test period divided by the number of active bins during the test period. Values of this measure greater than 2.5 g/bin (less than 10 individual observations) were excluded as outliers, probably the result of the monkey dumping out most of the food at once and then eating rather than exploring the puzzle. For one monkey, due to a programming error, counts were collected in 60 s bins on the first test day, so activity count data from this monkey were discarded for this day. One monkey was subsequently found to be blind and so data from this monkey were also discarded.

Data were analysed with linear mixed models in R v. 3.5.2 [19] using packages lme4 [20] and lmerTest [21]. Dependent variables were modelled with age (continuous) and test day (first or second) and their interaction as fixed factors and monkey as a random factor. Type III sums of squares were evaluated for statistical significance. Based on the suggestion of a reviewer of the Stage 2 submission of this report, we also evaluated models with random slopes for test day (but not intercepts, because models with both random slopes and intercepts were underidentified).

Based on the findings of Almeling and colleagues [11] demonstrating that animals over 19 years of age were unable to retrieve food awards and that animals over 24 years of age did not manipulate the objects, we completed an additional set of analyses to evaluate food retrieval and manipulation of the puzzle in animals younger and older than 19 (retrieval) and animals younger and older than 24 (manipulation). To that end, we considered a monkey to have successfully retrieved the reward if he or she obtained greater than 0 g of reward from the puzzle on either test day. Successful versus unsuccessful days were scored 1/0 and analysed with a Chi-square test comparing monkeys younger than 19 years of age to monkeys older than 19 years of age. Similarly, we evaluated the number of monkeys who did not manipulate the puzzle within 2 min and 20 min relative to their ages in order to determine if monkeys older than 24 did not manipulate the puzzle.

## 3. Results

Contrary to the findings of Almeling et al. [11], there were no effects of age on exploration, whether measured by the number of active bins out of a total of eight 15 s bins (figure 3a) or the total activity across the 2 min period (figure 3b). There were also no differences in exploration during the first 2 min between the 2 days on either measure. For number of active bins during the first 2 min, effect of age $F_{(1,\sim344.47)} = 0.31$, $p = 0.58$, day $F_{(1,\sim241.2)} = 0.0089$, $p = 0.92$, or age × day interaction $F_{(1,\sim240.8)} = 1.10$, $p = 0.29$. For total activity counts during the first 2 min, effect of age $F_{(1,\sim337.3)} = 0.17$, $p = 0.68$, effect of day $F_{(1,\sim241.8)} = 0.04$, $p = 0.84$, age × day interaction $F_{(1,\sim241.4)} = 1.16$, $p = 0.28$.

For the entire 20 min test period of each day, exploration tended to be higher on the second day compared to the first. For the total activity counts measure, this increased exploration appeared to be greater in the younger monkeys, yielding an interaction of age and day for this measure, albeit one that reflected an increase in exploration between days 1 and 2 in the younger monkeys

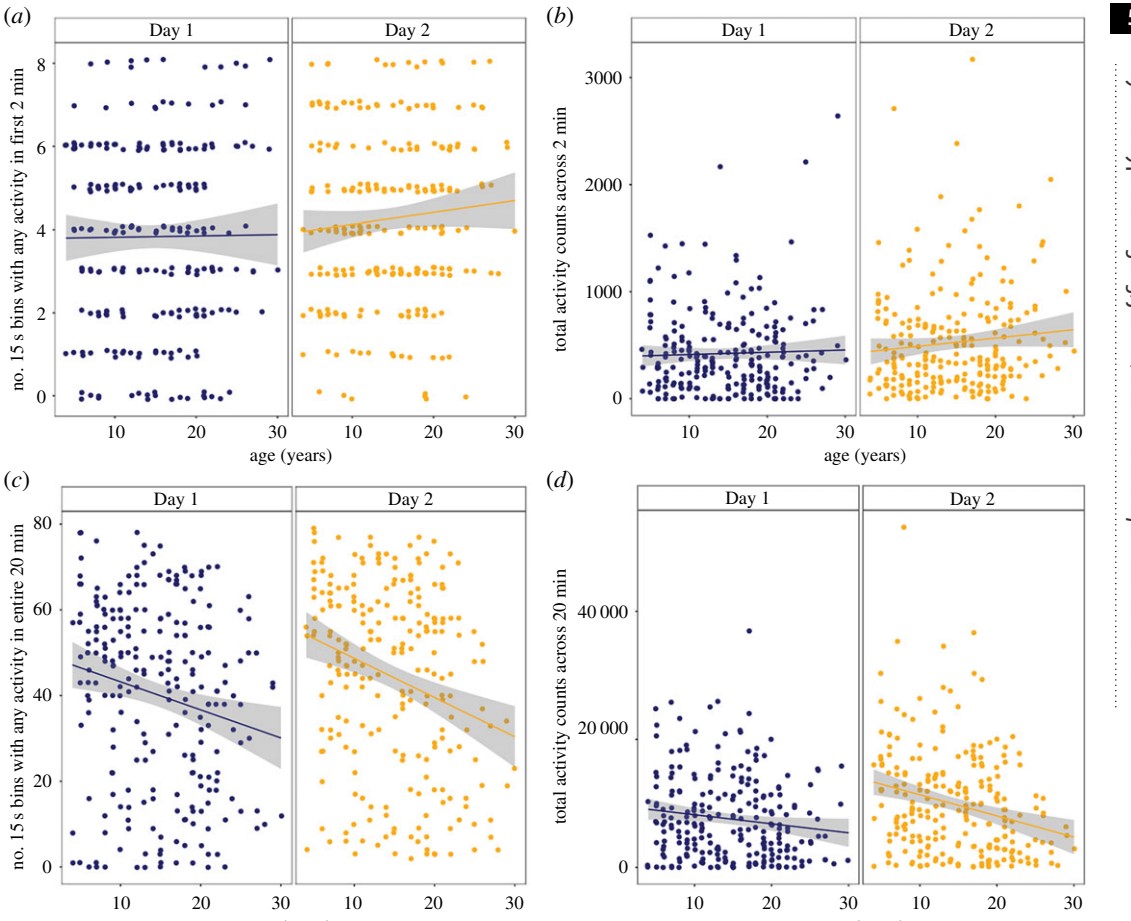

**Figure 3.** Exploration of the shaker as a function of age. Exploration metrics across 2 min (*a*,*b*) and the entire 20 min (*c*,*d*) test periods with Day 1 (blue) depicted separately from Day 2 (gold). (*a*,*c*) (left) depicts the number of active bins per each time duration by animal age. (*b*,*d*) (right) depicts the total activity counts per each time duration by animal age.

compared to the older rather than a loss of interest on the part of the older monkeys. For number of active bins during the 20 min (figure 3*c*), effect of age $F_{(1,\sim403.1)} = 1.33$, $p = 0.25$, effect of day $F_{(1,\sim239.9)} = 5.27$, $p = 0.023$, age × day interaction $F_{(1,\sim239.6)} = 1.26$, $p = 0.26$. For total activity counts during the entire 20 min period (figure 3*d*), effect of age $F_{(1,\sim433.6)} = 0.13$, $p = 0.72$, effect of day $F_{(1,\sim240.8)} = 23.4$, $p < 0.0005$, age × day interaction $= F_{(1,\sim240.6)}$, $p = 0.0044$. Inspection of the plots of these data suggested a possible extreme value on each day; excluding these two data points had no impact on the pattern of results.

We also analysed the mass of reward that monkeys were able to retrieve from the puzzle (figure 4). There were patterns suggesting that older monkeys might retrieve less food from the puzzle and all monkeys might retrieve more food from the puzzle on the second day compared to the first, but these did not reach statistical significance; effect of age $F_{(1,\sim432.1)} = 2.74$, $p = 0.098$, effect of day $F_{(1,\sim241)} = 3.41$, $p = 0.066$, age × day interaction $F_{(1,\sim241)} = 0.087$, $p = 0.77$. For the 'efficiency' measure of amount of food retrieved per active bin, there were no effects of age or suggestion that monkeys improved their efficiency in retrieving food on the second day compared to the first; effect of age $F_{(1,\sim348.2)} = 1.44$, $p = 0.23$, effect of day $F_{(1,\sim223.2)} = 0.32$, $p = 0.57$, age × day interaction $F_{(1,\sim222.7)} = 0.12$, $p = 0.73$. All of the preceding analyses produced essentially identical results when computed with random slopes instead of random intercepts.

Almeling *et al.* [11] found that animals over 19 years of age were unable to retrieve the reward and thus we tested whether monkeys over age 19 failed to retrieve the reward from the food retrieval puzzle. We considered a monkey to have successfully retrieved the reward if it obtained greater than 0 g of reward from the puzzle on either test day. We ran a chi-squared test of independence on frequencies of success and failure between monkeys 19 and younger compared to monkeys over 19. Fifteen of 183 monkeys that were 19 years of age and younger failed to retrieve any reward (8.2%),

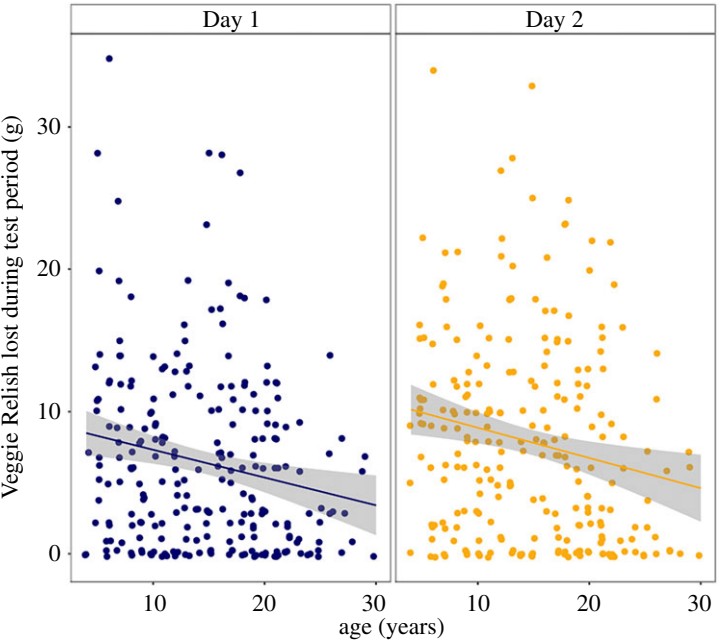

**Figure 4.** Amount of food reward retrieved from the shaker as a function of age. Total mass of reward freed from shaker during 20 min period for Day 1 (blue) and Day 2 (gold).

and 8 of 60 monkeys over 19 years of age (13.3%) failed to retrieve any reward. The chi-squared test of independence was not significant, $\chi^2_1 = 0.856$, $p = 0.355$, indicating that monkeys over 19 were not more likely to fail to obtain a reward from the puzzle.

Almeling *et al.* [11] also found that manipulation of the object was absent for animals over 24. On Test Day 1 in our study, eight monkeys did not manipulate the puzzle within 20 min; these monkeys were 6, 7, 8, 12, 13, 15, 21, 23 years of age (all younger than 24 years). On Test Day 2, all monkeys manipulated the puzzle within 20 min. If we limited our analysis to the first 2 min period (as in [11]), 20 monkeys did not manipulate the puzzle within 2 min on the first test day (mean age 15.2 years, ages 6, 6, 7, 8, 8, 12, 13, 14, 14, 15, 15, 17, 18, 19, 21, 21, 21, 22, 23, 24 years) and 7 did not manipulate on the second test day (mean age 15.14 years, ages 6, 9, 10, 19, 19, 19, 24). Thus, we found no evidence that monkeys over the age of 24 were any more likely to fail to manipulate the puzzle.

## 4. Discussion

Across a large sample ($N = 243$) of rhesus monkeys aged 4–30 years, we found no indication that monkeys lost interest in the 'non-social environment' with increased age [11], tested by the exploration of a novel food retrieval puzzle as in the prior study. There was no impact of age on animals' physical manipulation of the puzzle either over the first 2 min or the entire 20 min of the test period, or when the volume of food reward freed from the puzzle was considered. Notably, we used activity monitors to evaluate physical manipulation of the objects, rather than quantifying time touching the objects from video, in order to also capture potential differences in how vigorous physical activity was. This measure had the added benefit of eliminating any possible impact of observer bias regarding the age of the subjects on video scoring of exploration and manipulation. Previous evidence from human social psychology demonstrates that people hold strong beliefs and stereotypes about aged individuals [22,23] and human's attitude evaluations of aged non-human primates appears to be consistent with these beliefs [24].

Our findings challenge the notion that interest in the 'non-social world' declines with age in macaque monkeys, generally. The genus *Macaca*, the genus to which both Barbary and rhesus belong, includes 22 species whose behaviour along the tolerance–despotism continuum varies significantly resulting in different patterns of dominance and submission structures and behaviours, despite sharing a number of core biobehavioural features including living in matrilineal multi-male, multi-female groups with similar reproductive and developmental timing trajectories and fairly similar

behavioural repertoires [25]. Rhesus monkeys are significantly more despotic than the more-tolerant Barbary monkeys [25]. One possibility is that age-related changes in behaviour vary across species based on these fundamental social differences. Indeed, rhesus and Barbary monkeys displayed developmental differences in the same gaze following procedure [26]. When group-living rhesus and Barbary monkeys were tested in a procedure to evaluate the extent to which they tracked the gaze of a human experimenter, there was a subtle, but not significant decrease, in Barbary monkeys' tracking of gaze across age whereas rhesus monkeys evidenced a significant decrease in gaze tracking across age [26].

Housing conditions that affect daily activity budgets may also have impacted the variation in findings between Almeling and colleagues' and our study. Monkeys studied in the Almeling and colleagues study [11] lived outdoors in a primate park in social groups, whereas our monkeys lived indoors, typically in pairs. One possibility is that our monkeys were more inclined to manipulate objects because their living environments included less environmental or social stimulation. Given these issues, we do not view our findings as a strict failure to replicate those of Almeling *et al.* [11], but as suggesting that the phenomenon they reported may not generalize across species or context. Taking variation related to species, socialization history and testing context seriously may shed light not only on important evolutionary differences in socio-affective processes but also on factors that drive reproducibility in non-human primate science. As such, the view that positive social and affective experiences are prioritized with age as a theory to explain maintained social interest in aged humans (for a review [27]) remains an open possibility, as evidence for an evolutionarily conserved loss of non-social interest with ageing in monkeys is now mixed.

Some evidence exists suggesting that social interest might decline with age in rhesus monkeys. Aged monkeys show a reduction in gaze following of human experimenters [26] and a reduction in absolute levels of attention to photographs of other rhesus monkeys with ageing [28]. Thus, the impact of ageing on interest to both social and non-social aspects of the environment may vary depending on the manner in which these capacities are tested. Importantly, none of the existing studies (the current study, [11,24,25]) tested trade-offs between social and non-social processing by evaluating them within the same trial. In the face of greater limitations of physical and cognitive resources, elderly humans are thought to place more value on social relationships because of an awareness of one's own mortality and shorter remaining lifespan [27], motivations that non-human animals presumably do not possess. As such, it would be of interest to test monkeys across the lifespan in situations where they have the option of pursuing social interaction or viewing social stimuli, compared to interacting with novel objects that have the possibility of appealing food rewards.

Our findings suggest that there are no significant age-related changes in environmental exploration in rhesus monkeys and underscore the importance of studying ageing across multiple species in order to understand species nomothetic versus species-specific effects. Regardless of whether the current work is viewed as a conceptual replication of Almeling *et al.* [11] or not, what is clear is that greater attention to how ageing impacts variation in social and affective processes (including environmental exploration) is warranted. Decades of study of non-human primate models of cognitive ageing have shed light on similarities between ageing humans and ageing monkeys in trajectories of cognitive function and the neurobiology of cognition [29,30]. These studies also provide insight into the divergence between 'normal ageing' and age-related diseases in humans, as well as the ways in which ageing may contribute to risk for diseases of cognition; as such, they provide an opportunity to understand human and non-human animal ageing-related health and disease and to develop interventions to promote well-being. The mechanisms that support the significant changes that human affective life undergoes with age can and should be investigated in the same way.

Animal ethics statement. All procedures were approved by the University of California Davis Institutional Animal Care and Use Committee. The current protocol number is #20107.

Data accessibility. Anonymized data and the R code used to analyse the data have been deposited into Dryad Digital Repository: https://doi.org/10.5061/dryad.1bj133v [31].

Authors' contributions. Experimental design, data analysis, interpretation of data, drafting and editing article, accountability for all aspects of work: E.B.M. and M.G.B. Supervision of data collection, ethics approval, equipment source: E.B.M.

Competing interests. Neither author has competing interests.

Funding. E.B.M. and M.G.B.: P01AG016765, E.B.M.: University of California, Davis.

Acknowledgements. Thank you to Hailey Caparella-Veal, Ashley Murphy and Gilda Moadab for data collection. Thank you to the staff of Behavioral Health Services at the California National Primate Research Center for designing the puzzles.

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
