## [Reviewer comments · Royal Society Open Science]

Review History

RSOS-182237.R0 (Original submission)

Review form: Reviewer 1 (Julia Fischer)

Do you have any ethical concerns with this paper?

No

Have you any concerns about statistical analyses in this paper?

No

Recommendation?

Accept in principle

Comments to the Author(s)

I am very happy to see this conceptual replication of (parts of) our study, and I am eager to see the results. A minor issue: the hypotheses could be sharpened, and there is a typo on page 4, line 14 (double insertion of reference numbers). Regarding the discussion, I am sure that the authors will not only consider species differences but also substantial differences in housing conditions,

with the Barbary macaques living in a large 20ha enclosure in near natural social groups while the rhesus monkeys in this study have little else to do. Yet, this does not constitute a criticism, just the encouragement to consider this in the discussion of variation in motivation in the two study populations. Best wishes, Julia Fischer

Review form: Reviewer 2

Do you have any ethical concerns with this paper?

No

Have you any concerns about statistical analyses in this paper?

No

Recommendation?

Reject

Comments to the Author(s)

This study by Bliss-Moreau and Baxter addresses the interest in manipulating a baited puzzle as well as the success rate in Rhesus macaques across adult ages. The sample size is substantial (N=243) and with the described statistical approach valid results were obtained. Accordingly, the study has the potential to contribute to our understanding of cognitive aging processes in NHPs regarding exploration, persistence and learning (speed).

For a number of reason I think this study is not suited to be published in the section "Replications".

The authors state that they carried out a "conceptual replication" of a small part of a previous study from Almeling and colleagues - namely of whether old rhesus monkeys loose interest in nonsocial stimuli.

The journal, however, commits to publishing close replication attempts

Among others, these are the main reason of why I think this study is not a close replication (of a part of the original study):

1) The task is highly different: In the current study the monkeys have to rotate the top over the bottom of the puzzle to be rewarded; several portions of rewards can be obtained within one trial. The monkeys were exposed to the task for 20 min and tested on two days. In the original study monkeys were tested only one time for approximately 2min in a problem-solving task involving food (plus two sorts of toys (not involving food)) and rewarded only one time in case they poked through paper tissue at the end of a tube and turned it so that the reward fell out. The reward obtained after one appropriate move by the monkey in the current study, may motivate subjects subsequent engagement with the puzzle, which I think is a crucial point in addition to the difference to the task as such.

The authors put some effort in making the analysis comparable to the one in the original study, e.g. they analyze monkeys exploratory tendency for the first two minutes (in addition to the analysis of the behavior in 20 min), and also they determine success on a yes/no basis (with more than 0 gram of reward obtained during 20 min trial counting as success)

This is reasonable, taken that the authors aim to publish in the "Replication" section - but I do not feel it helps given the differences in the task. Further, I think this is unnecessary repeated testing of the same hypothesis.

2) Rearing conditions and species: The rhesus monkeys in the current study live in indoor cages either alone or in pairs, while in the original study Barbary macaque subjects lived in an outdoor enclosure in groups resembling rather natural conditions in terms of social life.

Given the absence of any assessments of social behavior and interest in social information the authors refer to a recent study of Rosati and colleagues and state that this one would “replicate” the effect of “sustained interest in social information by conspecifics (i.e., facial behaviors)” (Introduction page 3 l.43)

- this not correct.

Almeling et al did not find an effect of age on interest in playbacks of screams, photograph stimuli of neutral faces of conspecifics and frequency of “vocal commenting” on interactions of conspecifics.

Rosati and colleagues addressed the question of whether Rhesus macaques have an increasing preferential interest with age for a) Positive over neutral and b) Neutral over negative facial expression and found evidence for a negativity bias with age (which is in contrast to previous observations in humans). In both of their experiments their analyses revealed that younger monkeys showed greater interest in the photographic stimuli than older monkeys.

As a whole, I appreciate the experimental design and study and would suggest the authors to submit in the “Article” section (focusing on cognitive aspects of aging rather than relating the study to social and socio-emotional aspects of aging).

Decision letter (RSOS-182237.R0)

06-Feb-2019

Dear Dr Bliss-Moreau,

The Editors assigned to your Stage 1 Replication submission ("Interest in nonsocial novel stimuli as a function of age in rhesus monkeys") have now received comments from reviewers. We would like you to revise your paper in accordance with the referee and editors suggestions which can be found below (not including confidential reports to the Editor). Please note this decision does not guarantee eventual acceptance.

Please submit a copy of your revised paper within three weeks (i.e. by the The author due date is unavailable). If deemed necessary by the Editors, your manuscript will be sent back to one or more of the original reviewers for assessment. If the original reviewers are not available we may invite new reviewers.

When submitting your revised manuscript, you must respond to the comments made by the referees and upload a file "Response to Referees" in the "File Upload" step. Please use this to document how you have responded to the comments, and the adjustments you have made. In order to expedite the processing of the revised manuscript, please be as specific as possible in your response.

Once again, thank you for submitting your manuscript to Royal Society Open Science and I look forward to receiving your revision. If you have any questions at all, please do not hesitate to get in touch. Full author guidelines may be found at <http://rsos.royalsocietypublishing.org/page/replication-studies#AuthorsGuidance>.

Kind regards,
Professor Chris Chambers
Royal Society Open Science
openscience@royalsociety.org

on behalf of Chris Chambers (Registered Reports Editor, Royal Society Open Science)
openscience@royalsociety.org

Associate Editor Comments to Author (Professor Chris Chambers):

Two expert reviewers have now appraised the manuscript. The reviewers differ markedly in their overall recommendations, with one recommending acceptance in principle (Reviewer 1) and the other recommending reject (Reviewer 2). The main objection of Reviewer 2 is that the study deviated substantially from the methodology of the original study and therefore fails to meet Stage 1 Criterion 2: "Whether the manuscript describes a sufficiently valid (i.e. close)...replication of the original study methods and rationale to provide an indication of replicability." I think there is merit in this criticism, and the authors may well agree given their labelling of the study as a conceptual replication. However, I would like to hear the authors' rebuttal on this point before making a final judgment.

Given the lack of other substantive criticisms, I will not reject the submission solely on the basis of the deviation from the original methodology. Instead, following the authors' response (and further editorial consideration), should the study be deemed to have fallen short of meeting Stage 1 Criterion 2, and should all other primary criteria be met, I will offer the authors the opportunity to publish the article as either a regular article or a Registered Report, where the condition of close replication does not apply. In this case the article would be offered in-principle acceptance via the Replications track, and would proceed as normal through the current workflow, but if/when accepted at Stage 2 would appear in either the Research Article or Registered Reports category.

Comments to Author:
Reviewer: 1

Comments to the Author(s)

I am very happy to see this conceptual replication of (parts of) our study, and I am eager to see the results. A minor issue: the hypotheses could be sharpened, and there is a typo on page 4, line 14 (double insertion of reference numbers). Regarding the discussion, I am sure that the authors will not only consider species differences but also substantial differences in housing conditions, with the Barbary macaques living in a large 20ha enclosure in near natural social groups while the rhesus monkeys in this study have little else to do. Yet, this does not constitute a criticism, just the encouragement to consider this in the discussion of variation in motivation in the two study populations. Best wishes, Julia Fischer

Reviewer: 2

Comments to the Author(s)

This study by Bliss-Moreau and Baxter addresses the interest in manipulating a baited puzzle as well as the success rate in Rhesus macaques across adult ages. The sample size is substantial (N=243) and with the described statistical approach valid results were obtained. Accordingly, the study has the potential to contribute to our understanding of cognitive aging processes in NHPs regarding exploration, persistence and learning (speed).

For a number of reason I think this study is not suited to be published in the section "Replications".

The authors state that they carried out a "conceptual replication" of a small part of a previous study from Almeling and colleagues – namely of whether old rhesus monkeys loose interest in nonsocial stimuli.

The journal, however, commits to publishing close replication attempts

Among others, these are the main reason of why I think this study is not a close replication (of a part of the original study):

1) The task is highly different: In the current study the monkeys have to rotate the top over the bottom of the puzzle to be rewarded; several portions of rewards can be obtained within one trial. The monkeys were exposed to the task for 20 min and tested on two days. In the original study monkeys were tested only one time for approximately 2min in a problem-solving task involving food (plus two sorts of toys (not involving food)) and rewarded only one time in case they poked through paper tissue at the end of a tube and turned it so that the reward fell out.

The reward obtained after one appropriate move by the monkey in the current study, may motivate subjects subsequent engagement with the puzzle, which I think is a crucial point in addition to the difference to the task as such.

The authors put some effort in making the analysis comparable to the one in the original study, e.g. they analyze monkeys exploratory tendency for the first two minutes (in addition to the analysis of the behavior in 20 min), and also they determine success on a yes/no basis (with more than 0 gram of reward obtained during 20 min trial counting as success)

This is reasonable, taken that the authors aim to publish in the "Replication" section – but I do not feel it helps given the differences in the task. Further, I think this is unnecessary repeated testing of the same hypothesis.

2) Rearing conditions and species: The rhesus monkeys in the current study live in indoor cages either alone or in pairs, while in the original study Barbary macaque subjects lived in an outdoor enclosure in groups resembling rather natural conditions in terms of social life.

Given the absence of any assessments of social behavior and interest in social information the authors refer to a recent study of Rosati and colleagues and state that this one would "replicate" the effect of "sustained interest in social information by conspecifics (i.e., facial behaviors)" (Introduction page 3 l.43)

- this not correct.

Almeling et al did not find an effect of age on interest in playbacks of screams, photograph stimuli of neutral faces of conspecifics and frequency of "vocal commenting" on interactions of conspecifics.

Rosati and colleagues addressed the question of whether Rhesus macaques have an increasing preferential interest with age for a) Positive over neutral and b) Neutral over negative facial expression and found evidence for a negativity bias with age (which is in contrast to previous observations in humans). In both of their experiments their analyses revealed that younger monkeys showed greater interest in the photographic stimuli than older monkeys.

As a whole, I appreciate the experimental design and study and would suggest the authors to submit in the "Article" section (focusing on cognitive aspects of aging rather than relating the study to social and socio-emotional aspects of aging).

Author's Response to Decision Letter for (RSOS-182237.R0)

See Appendix A.

Decision letter (RSOS-182237.R1)

20-Feb-2019

Dear Dr Bliss-Moreau

On behalf of the Editor, I am pleased to inform you that your Manuscript RSOS-182237.R1 entitled "Interest in nonsocial novel stimuli as a function of age in rhesus monkeys" has been accepted in principle for publication in Royal Society Open Science.

As discussed, the central concern at Stage 1 was whether your replication protocol was sufficiently similar to the target study to meet Stage 1 Criterion 2: "Whether the manuscript describes a sufficiently valid (i.e. close) and robust (e.g. statistically powerful) replication of the original study methods and rationale to provide an indication of replicability". After reading (and in some cases re-reading) your paper, the target paper, the reviews received of your initial submission, and your response, I have decided that this criterion is sufficiently met for the article to continue as a Replication (and so without being transferred to a different article type). To explain further, Stage 1 Criterion 2 provides room for editorial discretion to permit deviations from the original methodology; in particular, what counts as "sufficiently close" to "provide an indication of replicability" is a subjective judgment. I was convinced that the similarity in methodology between your study and the target study is sufficiently high to "provide an indication of replicability", and I am persuaded that the rarity of replications (of any kind) in this field maximises both the value of the work and the editorial tolerance for deviations.

You may now progress to Stage 2 and complete the manuscript as approved.

Please note that you must now register your approved protocol on the Open Science Framework (<https://osf.io/rr>), using the "Submit your approved Registered Report" option and then the "Registered Report Protocol Preregistration" option. Please use the Registered Report option even though your article is being accepted as a Stage 1 Replication. Further into the registration process, in the Journal Title field enter "Royal Society Open Science (Replication article type, Results-Blind track)". Please note that a time-stamped, independent registration of the protocol is mandatory under journal policy, and manuscripts that do not conform to this requirement cannot be considered at Stage 2. The protocol should be registered unchanged from its current approved state. Please include a URL to the protocol in your Stage 2 manuscript, and because you submitted via the Results-Blind track please note in the manuscript that the pre-registration was performed after data analysis (e.g. "This article received results-blind in-principle acceptance

(IPA) at Royal Society Open Science. Following IPA, the accepted Stage 1 version of the manuscript, not including results and discussion, was preregistered on the OSF (URL). This preregistration was performed after data analysis."

Following completion of your study, we invite you to resubmit your paper for peer review as a Stage 2 Replication. Please note that your manuscript can still be rejected for publication at Stage 2 if the Editors consider any of the following conditions to be met:

- The Introduction and methods deviated from the approved Stage 1 submission (required).
- The authors' conclusions were not considered justified given the data.

We encourage you to read the complete guidelines for authors concerning Stage 2 submissions at: <http://rsos.royalsocietypublishing.org/page/replication-studies#AuthorsGuidance>. Please especially note the requirements for data sharing and that withdrawing your manuscript will result in publication of a Withdrawn Registration.

Once again, thank you for submitting your manuscript to Royal Society Open Science and I look forward to receiving your Stage 2 submission. If you have any questions at all, please do not hesitate to get in touch. We look forward to hearing from you shortly with the anticipated submission date for your stage two manuscript.

Kind regards,
Professor Chris Chambers
Royal Society Open Science
openscience@royalsociety.org

Author's Response to Decision Letter for (RSOS-182237.R1)

See Appendix B.

RSOS-182237.R2 (Revision)

Review form: Reviewer 1 (Julia Fischer)

Do you have any ethical concerns with this paper?

No

Have you any concerns about statistical analyses in this paper?

Yes

Recommendation?

Accept with minor revision

Comments to the Author(s)

The results are pretty clear. I suggest however to rerun the analysis and include random slopes into the models. Not that this will greatly affect the results, but it is important to be correct. I was

a bit puzzled why the results presented in Figure 3C did not reveal any significant age effect, given that the CI of the young and the old individuals did not overlap, but I trust the authors that this result is correct. Once the stats have been re-run, the paper can be accepted. The discussion is appropriately cautious. Overall, this is an important contribution to the literature.

Julia Fischer

Decision letter (RSOS-182237.R2)

23-Jul-2019

Dear Dr Bliss-Moreau

On behalf of the Editor, I am pleased to inform you that your Stage 2 Replication submission RSOS-182237.R2 entitled "Interest in nonsocial novel stimuli as a function of age in rhesus monkeys" has been accepted for publication in Royal Society Open Science subject to minor revision in accordance with the referee suggestions. Please find the referees' comments at the end of this email.

The reviewers and Subject Editor have recommended publication, but also suggest some minor revisions to your manuscript. Therefore, I invite you to respond to the comments and revise your manuscript.

Please also ensure that all the below editorial sections are included where appropriate (a non-exhaustive example is included in an attachment):

- Ethics statement

- Data accessibility

If you wish to submit your supporting data or code to Dryad (<http://datadryad.org/>), or modify your current submission to dryad, please use the following link:
<http://datadryad.org/submit?journalID=RSOS&manu=RSOS-182237.R2>

- Competing interests

- Authors' contributions

- Acknowledgements

- Funding statement

Because the schedule for publication is very tight, it is a condition of publication that you submit the revised version of your manuscript within 7 days (i.e. by the 31-Jul-2019). If you do not think you will be able to meet this date please let me know immediately.

- 1) A text file of the manuscript (tex, txt, rtf, docx or doc), references, tables (including captions) and figure captions. Do not upload a PDF as your "Main Document".
- 2) A separate electronic file of each figure (EPS or print-quality PDF preferred (either format should be produced directly from original creation package), or original software format)
- 3) Included a 100 word media summary of your paper when requested at submission. Please ensure you have entered correct contact details (email, institution and telephone) in your user account
- 4) Included the raw data to support the claims made in your paper. You can either include your data as electronic supplementary material or upload to a repository and include the relevant DOI within your manuscript

5) Included your supplementary files in a format you are happy with (no line numbers, Vancouver referencing, track changes removed etc) as these files will NOT be edited in production

Kind regards,
Professor Chris Chambers
Royal Society Open Science
openscience@royalsociety.org

on behalf of Chris Chambers (Registered Reports Editor, Royal Society Open Science)
openscience@royalsociety.org

Editor Comments to Author (Professor Chris Chambers):

The Stage 2 manuscript was returned to both reviewers who assessed the manuscript at Stage 1. One reviewer was not available. Based on the available reviewer's assessment, and my own reading of the manuscript, I have decided to proceed without further in-depth review. As you will see, the reviewer is satisfied with the Stage 2 submission but recommends an additional analysis in the interests of robustness. Since this deviates from the approved methodology at Stage 1 (and deviations from analysis plans, especially, are unusual in Stage 2 Replications), I am not going to require it -- however I would encourage the authors to consider the suggestion carefully. The analysis could be reported briefly as an additional unregistered analysis or unregistered robustness check.

Reviewers' comments to Author:

Reviewer: 1

Comments to the Author(s)

The results are pretty clear. I suggest however to rerun the analysis and include random slopes into the models. Not that this will greatly affect the results, but it is important to be correct. I was a bit puzzled why the results presented in Figure 3C did not reveal any significant age effect, given that the CI of the young and the old individuals did not overlap, but I trust the authors that this result is correct. Once the stats have been re-run, the paper can be accepted. The discussion is appropriately cautious. Overall, this is an important contribution to the literature.

Julia Fischer

Author's Response to Decision Letter for (RSOS-182237.R2)

See Appendix C.

Decision letter (RSOS-182237.R3)

12-Aug-2019

Dear Dr Bliss-Moreau:

It is a pleasure to accept your Stage 2 Replication entitled, "Interest in nonsocial novel stimuli as a function of age in rhesus monkeys" in its current form for publication in Royal Society Open Science.

on behalf of Professor Chris Chambers (Subject Editor)
openscience@royalsociety.org

Appendix A

Editor's Comments

Two expert reviewers have now appraised the manuscript. The reviewers differ markedly in their overall recommendations, with one recommending acceptance in principle (Reviewer 1) and the other recommending reject (Reviewer 2). The main objection of Reviewer 2 is that the study deviated substantially from the methodology of the original study and therefore fails to meet Stage 1 Criterion 2: "Whether the manuscript describes a sufficiently valid (i.e. close)...replication of the original study methods and rationale to provide an indication of replicability." I think there is merit in this criticism, and the authors may well agree given their labelling of the study as a conceptual replication. However, I would like to hear the authors' rebuttal on this point before making a final judgment.

***Authors' Response.** As noted, we elected to call our study a "conceptual replication" on the basis of variation in method, study species, and testing context. There is no doubt but there is deviation from the original paper. Nonetheless, we do believe that the task at hand replicated the central tenant of the original paper by testing the same construct (exploration of a physical object to free a food reward) as it varied with the same independent variable (age). The criticism that the study is not a close enough replication to warrant consideration as a replication is fair but also one that is important to consider in the context of the precious resource that are captive nonhuman primates. Management practices (including social housing) differ significantly across facilities. Further, macaques as a genus are often considered, in the psychological and neuroscience literatures, as a unitary type, wherein data from anthropological and field studies point to differences in their social behavior and societal structures which may very well have an impact on psychological processes such as motivated action for rewards (as tested here). If the goal of replication science is to understand the extent to which a phenomenon is robust and emerges in multiple samples, then it is critical to test multiple species even if features like their housing are vary by and/or are constrained by different management practices. Further, because of the limited number of nonhuman primates that are available to participate in studies (especially aged nonhuman primates, are represented in the present study) and the significant challenges associated with working with them, the vast majority of nonhuman primate studies are **never** replicated, either exactly or conceptually. The potential impact of this is exacerbated by typically small sample sizes. Thus, we see conceptual replication of this nature - where we test the same construct as in the original paper in a very large sample - as an important step forward in understanding the evolution of psychological processes.*

We have added language that speaks to these points to the introduction.

Given the lack of other substantive criticisms, I will not reject the submission solely on the basis of the deviation from the original methodology. Instead, following the authors' response (and further editorial consideration), should the study be deemed to have fallen short of meeting Stage 1 Criterion 2, and should all other primary criteria be met, I will offer the authors the opportunity to publish the article as either a regular article or a Registered Report, where the condition of close replication does not apply. In this case the article would be offered in-principle acceptance via the Replications track, and would proceed as normal through the current workflow, but if/when accepted at Stage 2 would appear in either the Research Article or Registered Reports category.

***Authors' Response.** We would be happy to follow whatever process the editor feels is best. The study design and analysis were registered, opening up the possibility of the Registered Report. Given the fact that the study was a conceptual replication, we were keen to participate in the 'results blind' review process.*

Reviewer 1

Reviewer 1, Comment 1. A minor issue: the hypotheses could be sharpened,

Authors' Response to R1, C1. Thank you for this comment - we have added an explicit statement about hypotheses.

Reviewer 1, Comment 2. ..and there is a typo on page 4, line 14 (double insertion of reference numbers).

Authors' Response to R1, C2. Thank you for catching this; we have fixed it.

Reviewer 1, Comment 3. Regarding the discussion, I am sure that the authors will not only consider species differences but also substantial differences in housing conditions, with the Barbary macaques living in a large 20ha enclosure in near natural social groups while the rhesus monkeys in this study have little else to do.

Authors' Response to R1, C3. Yes, we will absolutely attend to these differences in the discussion. We elected to leave out discussion of these differences in the introduction because they might allude to the results that we found. As noted in the response to the editor, above, from our perspective, one of the major challenges with nonhuman primate work is that there are limited populations from which experimental samples can be drawn and they often differ in terms of species and living conditions. By carrying out conceptually similar work across species and living conditions, we believe that we can start to shed light on which psychological processes are species nomothetic, and how management practices may impact the pattern of results that emerges in psychological and neuroscience experiments.

Reviewer 1, Comment 4. Yet, this does not constitute a criticism, just the encouragement to consider this in the discussion of variation in motivation in the two study populations.

Authors' Response to R1, C4. Thank you very much, Professor Fischer. Your group's research has inspired ongoing, engaging conversation with both of our teams.

Reviewer 2

Reviewer 2, Comment 1. For a number of reason I think this study is not suited to be published in the section "Replications". The authors state that they carried out a "conceptual replication" of a small part of a previous study from Almeling and colleagues – namely of whether old rhesus monkeys loose interest in nonsocial stimuli.

Authors' Response to R2, C1. We term our work a conceptual replication with the specific intent to signal that there were differences between the methods and the species - but we did test the same psychological construct (i.e., exploration of a physical object to free a food reward) as it varied with the same independent variable (i.e., animal age).

Reviewer 2, Comment 2. The journal, however, commits to publishing close replication attempts. Among others, these are the main reason of why I think this study is not a close replication (of a part of the original study):

1) The task is highly different: In the current study the monkeys have to rotate the top over the bottom of the puzzle to be rewarded; several portions of rewards can be obtained within one trial. The monkeys were exposed to the task for 20 min and tested on two days. In the original study monkeys were tested only one time for approximately 2min in a problem-solving task involving food (plus two sorts of toys (not involving food)) and rewarded only one time in case they poked through paper tissue at the end of a tube and turned it so that the reward fell out.

The reward obtained after one appropriate move by the monkey in the current study, may motivate subjects subsequent engagement with the puzzle, which I think is a crucial point in addition to the difference to the task as such.

Authors' Response to R2, C2. We appreciate this comment and will address it more in the Discussion of the full version of the manuscript. This brings up an important motivational issue vis-a-vis the construct of interest in nonsocial stimuli.

Reviewer 2, Comment 3. The authors put some effort in making the analysis comparable to the one in the original study, e.g. they analyze monkeys exploratory tendency for the first two minutes (in addition to the analysis of the behavior in 20 min), and also they determine success on a yes/no basis (with more than 0 gram of reward obtained during 20 min trial counting as success). This is reasonable, taken that the authors aim to publish in the "Replication" section – but I do not feel it helps given the differences in the task. Further, I think this is unnecessary repeated testing of the same hypothesis.

Authors' Response to R2, C3. We appreciate this concern, although we do not share it. Because we collected additional data (two epochs of 20 minutes of exploration instead of a single epoch of 2 minutes), we allowed for the possibility that we might not see differences in the initial 2 minutes, but that differences would emerge with continued testing, perhaps in part due to differences in species and housing conditions as noted by both reviewers. Thus we think our study is strengthened, rather than weakened, by the inclusion of additional analyses.

Reviewer 2, Comment 4. 2) Rearing conditions and species: The rhesus monkeys in the current study live in indoor cages either alone or in pairs, while in the original study Barbary macaque subjects lived in an outdoor enclosure in groups resembling rather natural conditions in terms of social life.

Authors' Response to R2, C4. We think that variation in species and living conditions may provide interesting information about the extent to which processes are species general (or genus nomothetic) and consistent or inconsistent across social housing conditions. Given that macaques as a genus are often considered in the psychological and neuroscience literatures as a unified whole, and that housing conditions vary across facilities that house nonhuman primates, we believe that it is critical to carry out such work in order to understand the extent to which constructs are robust and replicable across samples.

Reviewer 2, Comment 5. Given the absence of any assessments of social behavior and interest in social information the authors refer to a recent study of Rosati and colleagues and state that this one would "replicate" the effect of "sustained interest in social information by conspecifics (i.e., facial behaviors)" (Introduction page 3 l.43)

– this not correct.

Almeling et al did not find an effect of age on interest in playbacks of screams, photograph stimuli of neutral faces of conspecifics and frequency of "vocal commenting" on interactions of conspecifics. Rosati and colleagues addressed the question of whether Rhesus macaques have an increasing preferential interest with age for a) Positive over neutral and b) Neutral over negative facial expression and found evidence for a negativity bias with age (which is in contrast to previous observations in humans). In both of their experiments their analyses revealed that younger monkeys showed greater interest in the photographic stimuli than older monkeys.

Authors' response to R2, C5: We apologize that we made an error in our haste to be succinct. The Rosati et al. study shows no shift of positivity bias with age (indeed, no positivity bias at all in rhesus monkeys) but more importantly a large main effect of age on absolute attention to the photographs, which is different to the findings of Almeling et al. We have removed the sentence suggesting that this study replicates Almeling et al. from the introduction and will instead consider this paper in the Discussion section of the full manuscript.

Reviewer 2, Comment 6. As a whole, I appreciate the experimental design and study and would suggest the authors to submit in the “Article” section (focusing on cognitive aspects of aging rather than relating the study to social and socio-emotional aspects of aging).

Authors' response to R2, C6. We're glad that the reviewer could see the value of our work, even if it did not meet the reviewer's criteria for a replication.

Appendix B

UNIVERSITY OF CALIFORNIA, DAVIS

BERKELEY • DAVIS • IRVINE • LOS ANGELES • MERCED • RIVERSIDE • SAN DIEGO • SAN FRANCISCO

SANTA BARBARA • SANTA CRUZ

DEPARTMENT OF PSYCHOLOGY
CALIFORNIA NATIONAL PRIMATE RESEARCH CENTER
UNIVERSITY OF CALIFORNIA DAVIS
ONE SHIELDS AVENUE
DAVIS, CALIFORNIA 95616

April 26, 2019

Dear Professor Chambers and Editorial Board of *Royal Society Open Science*,

I am writing to submit the Stage 2 completed manuscript '**Interest in nonsocial novel stimuli as a function of age in rhesus monkeys**' for consideration at *Royal Society Open Science* as a Registered Report.

This Stage 2 submission follows in-principle acceptance of the Stage 1 submission (RSOS-182237.R1). As specified in the acceptance letter, we have generated a new approved protocol on OSF following the Stage 1 acceptance, containing the accepted Stage 1 protocol: <https://osf.io/rnzs4/>

We will publicly archive all anonymized data and R code used for analysis upon Stage 2 acceptance. These can be provided to the editor and reviewers at this time if desired.

Thank you in advance for considering our Stage 2 submission.

Sincerely,

Eliza Bliss-Moreau, Ph.D.
Assistant Professor, Department of Psychology
Core Scientist, California National Primate Research Center
ebliismoreau@ucdavis.edu
530-752-6268

Appendix C

UNIVERSITY OF CALIFORNIA, DAVIS

BERKELEY • DAVIS • IRVINE • LOS ANGELES • MERCED • RIVERSIDE • SAN DIEGO • SAN FRANCISCO

SANTA BARBARA • SANTA CRUZ

DEPARTMENT OF PSYCHOLOGY
CALIFORNIA NATIONAL PRIMATE RESEARCH CENTER
UNIVERSITY OF CALIFORNIA DAVIS
ONE SHIELDS AVENUE
DAVIS, CALIFORNIA 95616

July 23, 2019

Dear Professor Chambers and Editorial Board of *Royal Society Open Science*,

Thank you and Dr. Fischer for the review of our paper, '**Interest in nonsocial novel stimuli as a function of age in rhesus monkeys**'.

We noted Dr. Fischer's concern about the statistical approach and have addressed it in the manuscript by adding the following text:

Based on the suggestion of a reviewer of the Stage 2 submission of this report, we also evaluated models with random slopes for test day (but not intercepts, because models with both random slopes and intercepts were underidentified).

We did evaluate the models with random slopes which produced the same pattern results; R files shared on Dryad have been updated to reflect that. That is noted on page 9 with the following text:

All of the preceding analyses produced essentially identical results when computed with random slopes instead of random intercepts.

The lack of overlap in the CIs that Dr. Fischer sites reflects the day X age interaction in which young monkeys are more interested in the objects on Day 2. The pattern is described in the text.

We hope our manuscript is now suitable for publication.

Sincerely,

Eliza Bliss-Moreau, Ph.D.
Associate Professor, Department of Psychology
Core Scientist, California National Primate Research Center
ebliissmoreau@ucdavis.edu
530-752-6268